# Carbon footprint driven deep learning model selection for medical imaging

**Raghavendra Selvan**[1,2]                                                RAGHAV@DI.KU.DK

[1] *Department of Computer Science &* [2] *Department of Neuroscience, Univ. of Copenhagen, Denmark*

## Abstract

Selecting task appropriate deep learning models is a resource intensive process; more so when working with large quantities of high dimensional data that are encountered in medical imaging. Model selection procedures that are primarily aimed at improving performance measures such as accuracy could become biased towards resource intensive models. In this work, we propose to inform and drive the model selection procedure using the carbon footprint of training deep learning models as a complementary measure along with other standard performance metrics. We experimentally demonstrate that increasing carbon footprint of large models might not necessarily translate into proportional performance gains, and suggest useful trade-offs to obtain resource efficient models.

**Keywords:** deep learning, carbon footprint, performance measures, segmentation

## 1. Introduction

Deep learning (DL) models continue to yield state-of-the-art performance in several domains, including in medical image analysis. In addition to the development of novel theoretical frameworks, the success of these models can be attributed to the availability of big data and powerful compute. This has led to wide ranging applications requiring increasingly larger models on bigger datasets. The underlying hyperparameter tuning of DL models makes them further more computationally demanding. As a consequence, the carbon footprint of model selection of DL models is also on the rise.

In recent work, the trend of increasing carbon footprint of DL has been highlighted in natural language processing (NLP) where some of the biggest models are being developed (Strubell et al., 2019; Henderson et al., 2020). Efforts towards sustainable DL with a focus on low resource utilisation is also being investigated with dedicated research tracks in academic conferences[1]. Similarly, initiatives are also being taken in the medical imaging community to tackle the environmental impact of resource intensive computational methods.[2] Furthermore, tools such as the *experiment-impact-tracker* (Henderson et al., 2020) and *Carbontracker* (Anthony et al., 2020) provide ways to passively monitor the carbon footprint of training DL models.

Model selection in DL is far more resource intensive process than during inference. This can be attributed to the cost of specifying suitable network architectures by optimising several hyperparameters. Model selection criteria are primarily aimed at improving some form of performance measure without taking the associated environmental costs. In this work, we propose to perform DL model selection that is driven by the training carbon footprint along with relevant performance measures. Using a class of DL models for medical image segmentation we investigate the trade-off between carbon footprint of training models and their segmentation performance. We use Carbontracker[3] (Anthony et al., 2020) to track and estimate the carbon footprint.

---

1. Green and Sustainable Natural Language Processing
2. OHBM Sustainability and Environment Action Special Interest Group
3. https://github.com/lfwa/carbontracker/

## 2. Experiments

The experiments were designed to highlight the trade-off between segmentation accuracy and the underlying carbon footprint of training DL based segmentation models.

**Data:** We use the LIDC-IDRI dataset[4] consisting of 1018 thoracic CT scans which are processed to obtain 15096 patches of size $128 \times 128$. Lesions present in these patches are annotated by four raters. We combine the four annotations to obtain a single union segmentation mask per patch. The dataset is split into training, validation and test splits in the ratio 60:20:20.

**Experiments:** We use the popular 2D U-net model in Ronneberger et al. (2015) of increasing complexity (due to number of parameters) achieved by varying the initial convolution filters in seven configurations $F = [4, 8, 16, 24, 32, 48, 64]$. All seven models were trained on an Nvidia RTX 3090 GPU with 24GB memory with a batch size of 128 (maximum possible for the largest model on this GPU), implemented in Py-

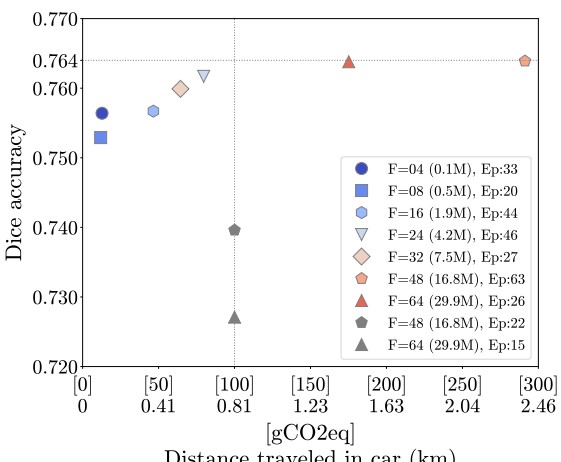

Figure 1: Test set accuracy of multiple U-net models of increasing complexity obtained with different initial filters $F = [4, 8, 16, 24, 32, 48, 64]$ along with the number of parameters and the epoch of convergence are shown. On the horizontal axis, the corresponding carbon footprint (in gCO2eq) and the equivalent distance traveled by car (in km) is reported. For a fixed carbon budget of 100gCO2eq the two most complex models do not converge and yield lower Dice accuracy (shown along the dotted line at 100gCO2eq).

Torch and optimised using the Adam optimiser using an initial learning rate of $5 \times 10^{-3}$. Models were trained to minimize Dice loss and segmentation accuracy was computed using the Dice coefficient. The carbon footprint was measured as the CO2 equivalent in weight (gCO2eq) of training the models, and the equivalent distance traveled by car (in km)[5] was estimated using Carbontracker.

**Results:** Figure 1 summarizes the results from our experiments where the test set segmentation accuracy of all the models are visualized along with their corresponding training carbon footprint. Unsurprisingly, the smaller of the models ($F = [4, 8]$) incur lower cost ($\approx$ 10gCO2eq) compared to the more complex ones ($F = [48, 64]$) which incur close to 20 times the cost (180 − 300gCO2eq). The segmentation accuracy does improve for the larger models compared to $F = [4, 8]$ models, by a small margin from 0.756 to 0.764 .

## 3. Discussion & Conclusions

The results in Figure 1 reveal some useful trade-offs when performing model selection. Firstly, the relative increase in carbon footprint between the best performing model (F=64) and the model with the smallest carbon footprint (F=4) is substantially more ($\approx$ 1500%) compared to the corresponding increase in Dice accuracy ($\approx$ 1%). Notice also that due to differences in convergence times, the most carbon intensive model is F=48 (Epoch:63)

---

4. https://wiki.cancerimagingarchive.net/display/Public/LIDC-IDRI

5. Carbontracker provides the conversion between gCO2eq to distance traveled by car based on the average CO2 emissions from newly registered motor vehicles in Europe in 2019

and not F=64 (Epoch:26). This indicates that penalizing larger models solely based on the number of parameters might not always be accurate.

In instances where access to compute resources is limited, practitioners could perform model selection by limiting the carbon budget. This is demonstrated by limiting the carbon budget to 100gCO2eq in Figure 1. Under this constraint the larger models (F=[48,64]) do not converge; when predicted using these models, obtained after exhausting the 100gCO2eq budget, the test accuracy is Dice=0.739 for F=48, Dice=0.727 for F=64 which are considerably lower than all other converged smaller models (for $F < 48$: $0.752 \leq \text{Dice} \leq 0.762$).

In this study we tracked the carbon footprint of models by running all the experiments. However, if carbon costs are to factor in model selection, then training all the models and choosing the model with the smallest carbon footprint is unreasonable. In such scenarios, tools like Carbontracker can be used to reliably predict the estimated carbon footprint by training the models for just a single epoch (Anthony et al., 2020). This can be used to constrain the architecture search to be within classes of models that fall within a specified carbon budget.

A natural extension to the idea of using carbon footprint for model selection is also to perform network architecture search (NAS) that is driven by the carbon footprint. This can be achieved by incorporating carbon emissions into the NAS minimization objective:

$$\hat{\theta} = \arg\min_{\theta} \left( \mathcal{L}_{\text{perf}}(\cdot; \theta) + \beta \cdot \mathcal{C}_{co2}(\cdot; \theta) \right), \tag{1}$$

where $\beta$ is a weighting factor. The optimal model parameter $\hat{\theta}$ minimizes the combined objective comprising performance measure $\mathcal{L}_{\text{perf}}(\cdot)$ and the carbon footprint of training the model $\mathcal{C}_{co2}(\cdot)$.

Another important consequence of using carbon footprint driven model selection is to provide fairer comparison of models. When comparing models that can be trained in parallel on large clusters, carbon footprint is a more balanced measure of complexity than other measures (like computation time) which can be reduced by scaling up resources.

In conclusion, we proposed to use carbon footprint as a complementary measure in DL model selection for medical image analysis. We demonstrated that in some scenarios the increase in computation complexity, and hence in carbon footprint of the model development, might not be justified relative to the gains in performance. We hope this paradigm for model selection will encourage practitioners to become aware of, and act on, the increasing carbon footprint leading to further research into resource efficient DL models.

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
