# OpenReview forum: "Carbon footprint driven deep learning model selection for medical imaging"
_MIDL.io/2021/Conference/Short — MIDL 2021 Poster_

### Official Review · Reviewer_zAWv · 2021-04-30

**Confidence:** 3
**Final Rating:** 3

**Summary:**

This paper focuses on the timely issue about environmental impacts of training bigger deep learning models and diminishing returns they may provide. Experiments on segmentation tasks show that small gains in performance using bigger models may lead to enormous costs in CO2 emissions. It also highlights the trade-offs appropriately and shows that the carbon footprint of training a model is a combination of convergence speed and parameters. Thus sometimes, bigger models can have small carbon footprints.


**Strengths:**

- This is a well-written discussion of an important issue.
- This paper provides good references for tracking carbon footprints for the experiments, which could be useful for other researchers in the community.
- The experiments presented are insightful.


**Weaknesses:**

- The experiments are limited to a few hyperparameters, mainly the width of the neural network. It would have been nice to compare different architectures for the task and other hyperparameters like the learning rate. I hope the author would expand on this in future versions.


**Deanonymize Review:**

no

**Detailed Comments:**

- The carbon footprint is reported by considering the epoch where the best performance is observed. Were the models trained with early stopping? If yes, what were the patience and other hyperparameters for training? More importantly, does figure 1 include these epochs (epochs spent after the best performance epoch)




**Justification Of The Rating:**

This paper has enough merits (good references, sufficient experiments, etc.) to be accepted as a short paper. The paper is well written; however, the experiments presented are limited. I would have preferred to give a 3.5 rating (accept) if there were an option for that.

**Paper Type:**

validation/application paper

**Special Issue:**

no

---

### Official Review · Reviewer_mDfB · 2021-05-04

**Confidence:** 4
**Final Rating:** 3

**Summary:**

The paper suggests driving model selection based on a combination of performance and training carbon footprint metrics. The paper demonstrates that the carbon footprint of increasingly more complex models does not lead to proportional gains in performance. The main goal of the paper seems to be to bring awareness to this finding.


**Strengths:**

- In general, given the increasing complexity of models in a variety of application areas, incorporating a penalty for carbon output is important to achieve sustainability.

- Experiments validate that often more compact networks can already achieve strong performance with a comparatively low carbon output. E.g. a U-Net with low parameterization achieves a Dice score of 0.756, whereas a higher parameterized U-Net only leads to a 0.01 gain but a 15-fold increase in carbon footprint for training.

- The paper also proposes several avenues of application – e.g. introducing a constraint on carbon footprint for model selection, adding a carbon-aware loss in neural architecture search.


**Weaknesses:**

- The *main weakness* of the paper that I see is that all of these conclusions seem generally applicable to all of deep learning. Within this submission they are simply also validated on a medical image segmentation problem. Also, I do not think it is *that* surprising that increasing parameterization leads to diminishing returns when it comes to performance. The carbon footprint naturally scales badly with increasing compute requirements.

- The paper would have significantly benefitted from arguments aimed specifically at medical imaging. For example, is it possible to estimate the carbon output of deep learning models with medical imaging applications within the whole DL field? A priori, I would assume that this percentage would be very low compared to other areas (e.g. NLP, the general Computer Vision field etc).

**Deanonymize Review:**

yes

**Justification Of The Rating:**

 Very weak accept. I vote for accepting the paper only because it brings awareness to a *very important issue within deep learning in general*. However, the paper lacks depth and justification when it comes to the importance of low carbon footprint models within medical imaging.


**Paper Type:**

both

**Special Issue:**

no

---

### Meta-Review · Area_Chair_Xg5h · 2021-05-06

**Recommendation:** Accept (Poster)
**Confidence:** 5

**Metareview:**

Both reviewers agree that while there are some weaknesses, the paper is generally sound and brings attention to an important and innovative issue.

---

### Decision · Program_Chairs · 2021-05-11

Accept (Poster)